# Strength, Stiffness, and Microstructure of Stabilized Marine Clay-Crushed Limestone Waste Blends: Insight on Characterization through Porosity-to-Cement Index

**DOI:** 10.3390/ma16144983

**Published:** 2023-07-13

**Authors:** Carlos Román Martínez, Yamid E. Nuñez de la Rosa, Daniela Estrada Luna, Jair Arrieta Baldovino, Giovani Jordi Bruschi

**Affiliations:** 1Applied Geotechnical Research Group, Department of Civil Engineering, Universidad de Cartagena, Cartagena de Indias 130015, Colombia; 2Faculty of Engineering and Basic Sciences, Fundación Universitaria Los Libertadores, Bogotá 110231, Colombia; 3Department of Mechanical Engineering, Federal University of Technology-Paraná, Curitiba 81280-340, Brazil; 4Department of Structural and Geotechnical Engineering, Polytechnic School, University of São Paulo, Ave. Prof. Luciano Gualberto, 380, Cidade Universitária, Butantã, São Paulo 05508-010, Brazil; 5Graduate Program in Civil Engineering, Universidade Federal do Rio Grande do Sul, Porto Alegre 90035-190, Brazil

**Keywords:** crushed limestone waste, porosity/cement index, unconfined compressive, stiffness, microstructure, empirical relationships

## Abstract

The porosity-to-cement index (η/C_iv_) has been extensively applied to study the evolution of different types of soil stabilization. However, this index has still not been used to characterize soils cemented with crushed limestone waste (CLW). In this sense, this paper sought to analyze the applicability of the porosity-to-cement index over the unconfined compressive strength ($q_{u}$) and initial stiffness at small deformations ($G_{o}$) of clayey soil improved with CLW and Portland cement. In addition, a microstructural analysis (SEM and EDX tests) was also conducted. CLW addition increased soil strength and stiffness over time. Moreover, $q_{u}$ and $G_{o}$ compacted mixtures containing CLW have established a distinctive correlation. Chemical microanalyses have uncovered a complex interfacial interaction between the soil, cement, and fine CLW particles, leading to a notable reduction in porosity.

## 1. Introduction

Cement production is widely recognized for its substantial environmental impact, with studies indicating that it accounts for a significant percentage of global carbon dioxide (CO_2_) emissions [1]. According to the International Energy Agency, cement production is responsible for approximately 8% of global CO_2_ emissions, making it one of the largest industrial sources of greenhouse gases [2]. Additionally, the process involves extensive resource consumption, including vast amounts of energy and raw materials. Moreover, the disposal of limestone waste poses an additional environmental challenge. Statistics from environmental agencies reveal that a significant portion of limestone waste generated by various industries remains underutilized and is often disposed of in landfills, exacerbating environmental concerns [3]. Addressing these issues through innovative approaches that reduce cement usage and promote the utilization of limestone waste is crucial for achieving sustainable development and mitigating the environmental impact of the construction industry [4].

The influence of rock powder on the geotechnical behavior of soils has been the subject of several studies. Investigations focus on the rock powder’s influence on organic soil’s behavior, aiming to enhance its engineering properties [5]. A separate experimental study examined expansive soil’s geotechnical properties and microstructure after stabilization with waste granite dust. The results provided insights into the effectiveness of waste granite dust as a stabilizing agent for expansive soils, demonstrating improvements in engineering behavior [6]. Additionally, the influence of rock powder on the geotechnical behavior of expansive soil was explored in another research endeavor. This study aimed to understand how rock powder affects the soil’s behavior and provided valuable insights into reducing swell potential and enhancing strength [7]. Combining the findings from these three studies, a comprehensive understanding of the geotechnical behavior and stabilization potential of soils with rock powder and waste granite dust can be achieved, leading to sustainable and effective soil stabilization practices in geotechnical engineering applications.

Concerning cemented soils, the relationship between porosity and cement content has been extensively explored in various studies investigating the properties of materials and stabilized soils. In the context of tailings disposal, the behavior of compacted filtered iron ore tailings−Portland cement blends has emerged as a new trend in Brazil [8]. This approach aims to utilize stacked disposal methods while evaluating the mechanical behavior of the materials. Additionally, the effect of fiber reinforcement on the mechanical behavior of reclaimed asphalt pavement-powdered rock−Portland cement mixtures has been investigated, shedding light on the potential for enhanced performance in these materials [9]. The strength and stiffness of compacted chalk putty−cement blends have also been studied to understand the behavior of this combination [10].

Furthermore, the mechanical response of fiber-reinforced reclaimed asphalt pavement sand−cement blends has been analyzed, providing insights into their structural performance [11]. Another study examined lime production residue glass powder binder-based geomaterials, examining their strength, durability, and microstructure [12]. A comprehensive study on the porosity/cement index, covering many porosities and cement contents, has contributed to a better understanding of the relationship between these parameters [13]. Lastly, the mechanical response of sand, reclaimed-asphalt pavement, and Portland cement mixture has been investigated, providing valuable insights into its behavior under different loading conditions [14]. These studies collectively highlight the significance of the porosity/cement relationship in assessing and optimizing the mechanical properties of various materials and stabilized soils. However, no studies have been conducted to establish the effectiveness of the porosity-to-binder ratio ($\eta /C_{\mathrm{iv}}$) for soil-limestone residue mixtures, nor a direct correlation between the stiffness of such mixtures and their initial stiffness at small deformations. Therefore, this article investigates the relationship between the $\eta /C_{\mathrm{iv}}$ index and unconfined compressive strength (UCS) and Go for soil-crushed limestone waste (CLW) mixes based on the previous studies about rock powder addition to soil stabilization (e.g., [9,15])

This study presents a pioneering investigation into the characterization of stabilized marine clay- CLW blends using a novel approach, the porosity-to-cement index. The blends’ strength, stiffness, and microstructure influence their performance in geotechnical applications. However, limited research has focused on systematically assessing these properties while considering the porosity-to-cement index. By leveraging this index, which offers a quantitative measure of the relationship between porosity and cement content, a comprehensive understanding of the material’s behavior can be achieved. This research aims to fill this critical gap in knowledge and provide valuable insights into the mechanical and microstructural characteristics of stabilized marine clay-CLW blends.

## 2. Experimental Program

The experimental program consisted of raw materials characterization (i.e., marine soil, cement, and CLW), molding and curing of specimens, UCS (q_u_) tests, stiffness (G_o_) measurement, and scanning electron microscopy (SEM-EDX) tests.

### 2.1. Materials

The soil samples were collected from the northern expansion area of Cartagena de Indias, Colombia, as described in the study by Baldovino et al. [16]. Clayey material was found in one of the soil slopes, and a comprehensive soil characterization was conducted. The characterization included granulometric analysis using a laser method with a dispersive solution-hexametaphosphate [17]. Atterberg limits were determined following the American Standard ASTM 4318 [18], and the specific gravity of soil particles was measured according to ASTM D854 [19]. The compaction characteristics were evaluated using the standard Proctor compaction curve under ASTM D698 [20]. The soil classification was performed based on the Unified Soil Classification System (USCS) as outlined in ASTM D2487 [21]. The granulometric curve of the soil sample is depicted in Figure 1, indicating that the soil sample consisted of 12% fine sand, 78% silt, and 10% clay, aligning with the classification from the Massachusetts Institute of Technology. The liquid limit of the soil was determined as 42%, while the plastic limit was 26.05%. Consequently, the plasticity index was calculated as 15.95%. The granulometric analysis yielded a uniformity coefficient (Cu) of 7.14 and a curvature coefficient (Cc) of 6.9 for the studied soil. The specific gravity determination resulted in an average value of 2.80 g·cm^−3^. Therefore, according to the USCS, the soil sample was classified as low-plasticity clay (CL). The chemical characterization of the soil was performed using an Oxford machine (Penta FET125 Precision) X-ACT in conjunction with a micro mass analyzer (LAMMA-1000 model X-ACT) and an X-ray energy dispersion spectrometer (XRD). Table 1 presents the chemical composition of the soil sample, which was primarily composed of silicon and aluminum. The XRD analysis detected the presence of kaolinite, quartz, and muscovite phases, exhibiting a highly crystalline behavior. The optimum moisture content of the soil was determined as 18%, and the maximum dry unit weight was measured as 17.60 kN·m^−3^, following the standard Proctor compaction test [20].

CLW was collected from a quarry near Cartagena de Indias. The granulometry of the material was not altered. Chemical composition, granulometry, and density tests were carried out. Figure 1 shows the granulometric curve of CLW, and the physical properties and chemical composition are summarized in Table 1 and Table 2, respectively. Figure 2 presents the SEM images of CLW. In addition, chemical composition of CLW is presented in Figure 3 through SEM-EDX microanalysis. Figure 3a show the SEM of CLW magnified at 20,000 times and Figure 3b presents the EDX chemical analysis of CLW detected region. It is possible to notice the non-spherical shape of the grains, surrounded by sharp edges, with a rough surface texture and localized points of high porosity. These characteristics contribute to a suitable CLW and soil cement anchorage.

The utilized cement was type III Portland cement (high initial strength), composed mainly of CaO (60.7%), MgO (4.1%), and SO_3_ (3.0%). The insoluble residue and the fineness (both in %) were measured as 0.77 and 0.04, respectively, in concordance with the manufacturer data. The axial strength at 28 days of curing is 53 MPa. Following the American Standard ASTM C150 [22], the specific gravity of cement grains was calculated as 3.11 g·cm^−3^. Cement was chosen as a binder due to its high use and manufacture in northern Colombia, where there are large natural deposits of limestone and local cement factories. Finally, distilled water was used for the characterization tests of the materials and molding of the specimens.

### 2.2. Specimens Molding and Preparation

Specimens molding and preparation were conducted in concordance to experimental program explained in Figure 4. Proctor standard compaction tests were carried out to define the molding points of the mixtures. The maximum dry unit weight was 17.6 kN·m^−3^ with an optimum moisture content of 18%. To study the influence of the initial porosity (inversely proportional to the dry unit weight) of the compacted blends on the strength and stiffness, two dry unit weights were chosen, one above the optimum point and another below. Thus, the following dry unit weights were selected: 18 kN·m^−3^ and 17 kN·m^−3^. Additionally, 18% moisture content was chosen as a fixed value for all molding points.

The percentages of cement varied from 3 to 6% and the CLW from 15 to 30%, considering previous studies on using other rock residues in new geomaterials [5,7,9]. Due to the rapid reaction of the cement, curing times of 7 and 28 days were selected.

### 2.3. UCS and Stiffness Program

Specimens intended for unconfined compression tests were prepared with precise dimensions of 100 mm in height and 50 mm in diameter. The preparation process encompassed several steps, including weighing, mixing, compacting, packaging, wet chamber storage, and curing. Each constituent, namely soil, CLW, water, and cement, was weighed with a high sensitivity of 0.01 g. The soil fraction was meticulously dried and sieved to ensure its proper condition. To achieve a homogeneous mixture, the additives were thoroughly mixed with the soil. The moisture content of each specimen was determined based on the total dry mass. The excess soil was then utilized to calculate and achieve the necessary moisture content of 18% (i.e., optimum water content) to facilitate the molding of triplicate specimens. Sample dimensions and compaction followed the indications of ASTM D1633 [23] and previous studies applied the same sample confection methodology [13,24].

Immediately after molding, the specimens were carefully placed in airtight bags to prevent any fluctuations in moisture content. They were then stored in a controlled wet chamber within the laboratory, where a temperature of 27 °C and a relative humidity of 95% were maintained, as recommended in recent studies [10,25]. To ensure the suitability of the specimens for testing, specific criteria were applied. These criteria included the degree of compaction, which had to be within ±1% of the target value; the moisture content, which had to be within ±0.5% of the target value; the diameter, which had to be within ±1% of the target value; and the height, which had to be within ±2% of the target value. Additionally, any deviations in diameter or height should never exceed 2 mm. The criteria for valid UCS/Go specimens were studied by Consoli et al. and Saldanha et al. [26,27,28].

After the curing period, before the mechanical testing, specimens were placed in water immersion to avoid the possible suction effects; although this process decreases suction, the influence is not entirely removed [29]. After immersion for 24 h, the specimens were subjected to Ultrasonic Pulse Velocity tests according to American standards ASTM C597 [30] and later to unconfined compression tests following the indications of American standards ASTM D1633 [23].

Several studies have related UCS (q_u_) to the porosity-to-cement index η/C_iv_ [13,25,31]. Most of these studies conclude that the volumetric content of cement (C_iv_) needs an adjustment of 0.28 to make q_u_ and C_iv_ compatible. This exponent depends on the type of soil and the type of binder used. The adjustment of $\eta /C_{iv}^{0.28}$ and q_u_ is done through a power relation, as shown in Equation (1).
(1)$$q_{u}=A{\left[\frac{\eta }{{\left(C_{iv}\right)}^{x}}\right]}^{-B}$$

The constant A in the Equation (1) is of significant theoretical importance as it depends on various factors that influence the mechanical behavior of the soil−cement mixture. These factors include the soil’s critical state strength ratio, the cement’s uniaxial compressive strength, the cement stress ratio, the porosity at a critical state, and the ratio between the uniaxial compression and extension strengths [32]. The exponents −B and x play a vital role in characterizing the relationship between the soil and cement properties and the strength behavior of the mixture. 

A thorough understanding of these parameters is crucial for accurately predicting the mechanical response of the stabilized material. The porosity, η, is the initial voids volume in relation to the total specimen volume. This study calculates the initial porosity based on the volumetric amounts of CLW, cement, and soil. Thus, this study is the first where the CLW variable is used within the porosity to determine the resistance and rigidity of soil-stabilized mixtures. Previous studies have inserted other variables into Equation (1), such as lime production residue [12], sugarcane bagasse ash [33], carbide lime [34], and glass residue [34]. Finally, the derivation of Equation (1) for three materials can be found in Baldovino et al. [35].

### 2.4. Microstructural Analysis

The scanning electron microscopy (SEM) technique is employed to capture a diverse range of electrons that are generated when an electron beam interacts with the surface of a sample. This interaction creates a detailed image of the sample’s features and composition. By scanning the emitted electrons, data are collected, resulting in an image that reflects the topography or composition of the sample. In addition, an energy-dispersive X-ray spectroscopy (EDX) device can be used to analyze the chemical composition of a specific area or point on the sample. The samples, namely Soil + 3% Cement + 15% CLW and Soil + 6% Cement + 30% CLW, underwent a curing process and unconfined compressive tests after 28 days of curing. To ensure reliable compositional analysis, the samples were polished using the wet method with ethanol to prevent material reactions and obtain a sufficiently smooth surface. Subsequently, the samples were vacuum dried and coated with a layer of gold for subsequent microscopy analysis. EDX compositional analyses were performed on selected points of the samples. Surface examinations of the compacted blends were conducted at magnifications of up to 2000 times, while magnifications of up to 5000 times were used to visualize the characteristics of hydrated calcium crystals. A LIRA3 TESCAN (Universidad de los Andes, Bogotá, Colombia) was used for conducting the SEM-EDX tests.

### 2.5. Statistical Analysis

The analysis of the mechanical results, considering the significance of the main effects, their interactions, and the verification of the presence of curvature in the response behavior, was carried out with the aid of statistical software applying an analysis of variance (ANOVA) at a significance level of 5% (*p*-value < 0.05). For the statistical analysis (univariate linear analysis), we utilized the SPSS Statistics Version 26 software. The objective of the ANOVA applied in this research was to determine whether there were significant differences between the means of the studied groups. This method was used to compare the means of four groups in order to assess whether any of the group means differed significantly from each other. In other words, the analysis aimed to identify meaningful variations in the dependent variables (UCS and initial shear stiffness) across different categories of independent variables (cement content, lime content, dry unit weight, and curing time).

## 3. Results and Discussions

### 3.1. Effects of Porosity/Cement Index and Curing Periods on Strength for Soil−CLW−Cement Blends

Aiming to combine the tests into a single relationship, the UCS results were related to the porosity/binder index (Figure 5 and Figure 6). This power fit led to equations that enabled the prediction of the strength of the cemented mixes for a wide range of porosities and cement concentrations. The binder content parameter’s influence on the response variable was adjusted using an exponent of 0.28; this exponent indicated the best fit for the mechanical results and was chosen after an iterative procedure on the studied data. The exponent’s positive value shows that the effects of porosity and binder content on mechanical characteristics are mutually exclusive.

The use of the exponent of 0.28 also depends on the empirical superposition of strength contributions of the soil and cement phase, the critical state, and the dependence of shear strength on the state parameter concepts, as well as other properties provided by Diambra et al. [32]. The value of 0.28 has been used to adjust not only the mechanical behavior of soil−cement mixtures [36,37] but also other types of materials, such as soil-carbide lime−ground glass mixtures [38], soil−cement−roof tile waste mixtures [39], lateritic soil−sand−cement compacted blends [25], and even mining tailings [40,41]. 

The parameter B (Equation (1)) is responsible for linking the peak strength to the state parameter, and its value depends on the specific characteristics of the soil and cement. On the other hand, the value of x is closely related to the value of B. Recent studies, along with Diambra’s theoretical derivation of artificially cemented granular soil, have shed light on the relationship between x and B, indicating that x can be expressed as the reciprocal of B [35,42,43,44]. This relationship suggests that, as B increases, x decreases, highlighting the influence of B on the overall strength characteristics. However, it is important to note that, in the present research, a slightly different relationship between x and B was observed. Specifically, the value of x was found to be 1.43 divided by B. This variation could be attributed to the specific experimental conditions, sample properties, or other influencing factors in the study. Further investigations and analyses are needed to fully comprehend the implications of this observed difference and its potential implications for the mechanical behavior of the soil−cement mixture.

Regarding the behavior of the mixture, it was observed that a decrease in the $\eta /C_{iv}^{0.28}$ index resulted in an increase in the UCS for all combinations. This finding suggests that the mechanical strength of the mixture is influenced by both porosity and cement content. When the porosity decreases, the friction within the soil mass increases, leading to improved interlocking and consequently enhancing the mechanical strength. Additionally, an increase in cement content promotes more cementitious reactions within the mixture, contributing to the development of a stronger soil−cement framework and consequently increasing the UCS. This physical-chemical phenomenon has been supported by various studies [37,45] that have demonstrated improved mechanical behavior of cemented geotechnical materials with a reduction in the porosity/binder content index. Furthermore, an increase in the content of calcined clay lightweight aggregate (CLW) also resulted in higher UCS. This behavior can be attributed to the coarser grain size distribution of the CLW material [25], which enhances the interlocking phenomenon within the mixtures, thereby further increasing the strength. However, it is important to note that the CLW particles functioned solely as an inert material in this context and did not engage in chemical interactions with the soft soil or the applied cement.

Regarding the effects of the curing period, strength values varied from 1.6 to 1.8 MPa for the 7 days and 2.2 to 2.6 MPa for the 28-day curing period. This indicates that the cementitious reactions continued to develop throughout the curing period. Studies of Baldovino et al. [35] have also evidenced increased cemented mixtures’ strength over more extended curing periods.

A good power fit (Equations (2)–(5)) between the UCS and η/C_iv_ index was evidenced for the mixtures, with determination coefficients (R^2^) of 0.85–0.98 and 0.97–0.98 for the 7-day and 28-day curing periods, respectively. This indicates the index viability to predict the UCS behavior of cemented CLW mixtures, also corroborated by studies applied to other geotechnical materials [12,46]. Since Equations (2) and (4) are adjusted to the same exponents, with only different scalars, the ratio between these scalars defines the variation magnitude of the UCS over the curing period (i.e., from 7 to 28 days, using 15% CLW), with a value of 37%.
(2)$$q_{u}=23.63\times 10^{9}{\left[\frac{\eta }{{\left(C_{\mathrm{iv}}\right)}^{0.28}}\right]}^{-5.11}(R^{2}=0.85)\;15\%\;\mathrm{CLW};\;7\text{-}d$$
(3)$$q_{u}=30.83\times 10^{6}{\left[\frac{\eta }{{\left(C_{\mathrm{iv}}\right)}^{0.28}}\right]}^{-5.11}(R^{2}=0.98)\;30\%\;\mathrm{CLW};\;7\text{-}d$$
(4)$$q_{u}=32.32\times 10^{9}{\left[\frac{\eta }{{\left(C_{\mathrm{iv}}\right)}^{0.28}}\right]}^{-5.11}(R^{2}=0.97)\;15\%\;\mathrm{CLW};\;28\text{-}d$$
(5)$$q_{u}=40.22\times 10^{9}{\left[\frac{\eta }{{\left(C_{\mathrm{iv}}\right)}^{0.28}}\right]}^{-5.11}(R^{2}=0.98)\;30\%\;\mathrm{CLW};\;28\text{-}d$$

### 3.2. Effects of Porosity/Cement Index and Curing Periods on Stiffness for Soil−CLW−Cement Blends

Figure 7 and Figure 8 present the initial shear stiffness at small deformations results for the cemented mixtures. Results were expressed as a function of the porosity/cement content index ($\eta /C_{iv}^{0.28}$). Once again, an exponent of 0.28 was utilized on the cement content parameter, being determined through iterative processes. Like the UCS results, porosity reduction and cement content increase (decreasing $\eta /C_{iv}^{0.28}$ values) led to an increment of Go, due to more significant friction mobilization and higher contact area between particles. Furthermore, similar observations have been documented regarding stabilizing various geotechnical materials. These studies have consistently shown that specimens with lower porosity and increased cement content exhibit elevated values of the coefficient of stiffness, Go. This trend holds across different materials and reinforces that higher cement content contributes to improved stabilization and enhanced mechanical properties [10,47]. As expected, the increase in the curing period improved the Go of all mixtures, indicating that the cementitious reactions continued to happen during this period; stiffness values varied from 7 to 8.5 MPa at 7 days and 8.6 to 10 MPa at 28 days. In addition, higher strength gains were evidenced for treatments of lower porosities and higher binder contents, considering that the increased precipitation of cementitious compounds contributed to strength development for these mixtures. Once again, increasing CLW content also increased the stiffness of the mixtures. The coarser grain size distribution of CLW enhanced the interlocking phenomenon, further increasing the stiffness. Moreover, CLW particles were evidenced as an inert material. 

Significant power fit were established between the coefficient of stiffness, Go, and the $\eta /C_{iv}^{0.28}$ index for both the 7-day and 28-day curing periods. The coefficient of determination (R^2^) values ranged from 0.87 to 0.96 for the 7-day curing period and from 0.91 to 0.97 for the 28-day curing period. These high R^2^ values indicate the effectiveness of the $\eta /C_{iv}^{0.28}$ index in accurately predicting the Go of CLW cemented soil mixtures. These findings are consistent with the studies conducted by Hoch et al. [10] and Bruschi et al. [41]. Equations (6)–(9) establish a power fit between the $\eta /C_{iv}^{0.28}$ index and the stiffness of the compacted blends for both the 7-day and 28-day curing periods, considering the use of 15% and 30% CLW content.
(6)$$G_{o}=36.13\times 10^{7}{\left[\frac{\eta }{{\left(C_{\mathrm{iv}}\right)}^{0.28}}\right]}^{-3.37}(R^{2}=0.87)\;15\%\;\mathrm{CLW};\;7\text{-}d$$
(7)$$G_{o}=44.22\times 10^{7}{\left[\frac{\eta }{{\left(C_{\mathrm{iv}}\right)}^{0.28}}\right]}^{-3.37}(R^{2}=0.96)\;30\%\;\mathrm{CLW};\;7\text{-}d$$
(8)$$G_{o}=44.44\times 10^{7}{\left[\frac{\eta }{{\left(C_{\mathrm{iv}}\right)}^{0.28}}\right]}^{-3.37}(R^{2}=0.91)\;15\%\;\mathrm{CLW};\;28\text{-}d$$
(9)$$G_{o}=51.83\times 10^{7}{\left[\frac{\eta }{{\left(C_{\mathrm{iv}}\right)}^{0.28}}\right]}^{-3.37}(R^{2}=0.97)\;30\%\;\mathrm{CLW};\;28\text{-}d$$

If Equations (6)–(9) are divided by Equations (2)–(5), respectively, a correlation between UCS and Go is calculated as presented by Equations (10)–(13). The Go/UCS relationship in concordance to Equations (10)–(13) is dependent on the CLW and porosity/cement index.
(10)$$\frac{G_{o}}{q_{u}}=15.29{\left[\frac{\eta }{{\left(C_{\mathrm{iv}}\right)}^{0.28}}\right]}^{-1.17}7\text{-}\mathrm{days},\;15\%\;\mathrm{CLW}$$
(11)$$\frac{G_{o}}{q_{u}}=14.84{\left[\frac{\eta }{{\left(C_{\mathrm{iv}}\right)}^{0.28}}\right]}^{-1.17}7\text{-}\mathrm{days},\;30\%\;\mathrm{CLW}$$
(12)$$\frac{G_{o}}{q_{u}}=13.75{\left[\frac{\eta }{{\left(C_{\mathrm{iv}}\right)}^{0.28}}\right]}^{-1.17}28\text{-}\mathrm{days},\;15\%\;\mathrm{CLW}$$
(13)$$\frac{G_{o}}{q_{u}}=12.87{\left[\frac{\eta }{{\left(C_{\mathrm{iv}}\right)}^{0.28}}\right]}^{-1.17}28\text{-}\mathrm{days},\;30\%\;\mathrm{CLW}$$

### 3.3. Effects of Porosity/Cement Index and Curing Periods on Stiffness for Soil−CLW−Cement Blends

Examining Equations (2)–(5) and (6)–(9) shows that the exponents outside the equations are identical. However, the initial constants associated with the UCS and Go results exhibit distinct values. Consoli et al. [48] suggested that the mechanical response of soil-cement mixtures using the same cement can be standardized by considering various predetermined factors such as curing time and temperature. Consequently, it is conceivable to develop a single mathematical correlation capable of predicting the mechanical behavior of the stabilized material. The relations expressed by Equations (2)–(9) can be normalized when divided by a particular value of resistance (UCS) or Go, which necessarily corresponds to a value $\eta /C_{iv}^{x}\;$= Δ. Therefore, we have a normalized value of q_u_ (q_u-n_) and Go (Go-n) for a given Δ, which results in the following generalized expressions (for UCS and Go, Equations (14) and (15)):(14)$$\frac{q_{u}}{q_{u-n}\left(\eta /C_{\mathrm{iv}}^{x}=\Delta \right)}=\frac{A{\left(\eta /C_{\mathrm{iv}}^{x}\right)}^{-B}}{A{\left(\Delta \right)}^{-B}}=\Delta^{B}{\left(\eta /C_{\mathrm{iv}}^{x}\right)}^{-B}$$
(15)$$\frac{Go}{Go-n\left(\eta /C_{\mathrm{iv}}^{x}=\Delta \right)}=\frac{A{\left(\eta /C_{\mathrm{iv}}^{x}\right)}^{-B}}{A{\left(\Delta \right)}^{-B}}=\Delta^{B}{\left(\eta /C_{\mathrm{iv}}^{x}\right)}^{-B}$$

Equations (14) and (15) allow the evaluation of the mechanical behavior and stiffness of cemented soil samples, within a spectrum of porosity and cementing agent contents, with only one value relative to the UCS and Go. If Δ=30, Equations (14) and (15) led to (Equations (16) and (17)):(16)$$\frac{q_{u}}{q_{u-n}\left(\eta /C_{\mathrm{iv}}^{0.28}=30\right)}=3,532,605.26{\left(\eta /C_{\mathrm{iv}}^{0.28}\right)}^{-5.11}(R^{2}=0.92)$$
(17)$$\frac{G_{o}}{G_{o-n}\left(\eta /C_{\mathrm{iv}}^{0.28}=30\right)}=95,038.29{\left(\eta /C_{\mathrm{iv}}^{0.28}\right)}^{-3.37}(R^{2}=0.92)$$

The determination coefficients found in Equations (16) and (17) were 0.92, which denotes the possibility of normalizing results even though two different curing periods were studied and CLW was employed in the compacted blends. Moreover, the elevated R^2^ indicates the equations’ great representativeness in predicting the analyzed mixtures’ strength and stiffness. Thus, Figure 9 presents the normalization of the UCS at $\eta /C_{iv}^{0.28}=30$ considering 15–30% CLW and 7–28 days of curing, and Figure 10 shows the normalization of stiffness at $\eta /C_{iv}^{0.28}=30$ considering 15–30% CLW and 7–28 days of curing. The superposition of calibrating the normalized UCS curve is plotted in Figure 10. Because there is an evident correlation between the UCS increase and the Go increase, in Figure 11, both results were plotted, and a linear correlation between strength and stiffness is evident, as found in recent studies [10,34,49].

### 3.4. Microstructure and Microanalysis of Compacted Blends

The microstructure of the mixtures was studied employing SEM-EDX microanalysis. Figure 12, Figure 13 and Figure 14 show the microstructure of a cemented soil mix, indicating that cement addition mainly resulted in the formation of C-S-H gels and ettringite crystals. No evidence was shown regarding the reaction of the limestone particles with the cement. As shown in Figure 12a and Figure 14a,b, CLW particles have not reacted but have formed a strong matrix with the foundation products, as no voids were seen around the particles. Replacing part of the Portland cement with limestone particles can provide additional surface for precipitation of hydration while also decreasing the water needed to keep the soil workable (Figure 12 and Figure 13). The medium and large CLW particles acted as physical anchors within the matrix, interlocking the soil aggregates and silicon films, hence withstanding the tensile stresses developed during desiccation (Figure 12b and Figure 14a).

Ettringite was formed from the reaction between calcium sulfate in cement and tricalcium aluminate (Figure 12 and Figure 13). Large Ettringite crystals can be showed in Figure 13. Adding CLW can increase the amount of sulfate in the mix and favor ettringite formation. Ettringite can contribute to improving the resistance and durability of cemented soil. The hydrated calcium silicate present in Figure 12, Figure 13 and Figure 14 was formed from the reaction between the tricalcium silicate present in the cement and the water in the mix. Limestone dust can increase the amount of silicate available in the mix and therefore favor the formation of C-S-H (Figure 12a and Figure 13b).

In general, the soil−cement microstructure encompasses the intricate arrangement of soil particles, cementitious matrix, and the interfacial transition zone. Within the composite, soil particles exhibit variability in size, shape, and composition, thereby substantially influencing the mechanical behavior and performance of soil−cement−CLW blends. The cementitious matrix, a result of the hydration process involving cement and water, comprises hydrated minerals that confer strength and cohesiveness to the composite. The interfacial transition zone, an essential component, represents the interface where soil particles interface with the cement paste (and form silica film). Its thickness, composition, and porosity significantly govern the compacted blends’ interparticle bonding, stress transfer mechanisms, and load-bearing capacity as presented in Figure 14. A comprehensive understanding and meticulous optimization of the microstructural characteristics of soil−cement−CLW mixtures are vital for achieving the desired strength, durability, and overall performance of the new geomaterial.

### 3.5. Statistical Analysis

Unconfined compressive and stiffness Go results were used to perform a statistical ANOVA which included the analysis of four control factors, cement content, CLW content, dry unit weight, and curing period. ANOVA results are shown in Table 3 and Table 4 for UCS and stiffness, respectively. Unconfined Compressive indicates that all studied factors and their second-order interactions significantly influence response variable *q*_u_. As for the influence of the main effects, the increase of all main factors increased the mechanical behavior and stiffness of the compacted blends. The behavior mentioned above further corroborates the results of the mechanical tests, indicating that cement content, CLW content, dry unit weight, and curing period play a fundamental role in the development of strength and stiffness. The ANOVA indicates that second-order interactions with CLW are insignificant for stiffness. This means there is not enough evidence to affirm that the combined effect of the CLW and the other variables differs from the sum of their individual effects.

On the other hand, if the CLW variable is not considered for the ANOVA, the cement, cure time, and dry unit weight variables are significant, as well as the second-order interactions between them.

The primary goal of the ANOVA was to determine if the observed differences between group means were due to genuine differences in the populations they represent or if they could be attributed to random chance. The second goal was to provide statistical evidence to support the mechanical results. Table 3 and Table 4 show that all four main factors significantly influence the response variables (represented by p-values below the 0.05 significance level) for the UCS and initial shear stiffness. Therefore, in any field or real-life applications, all the variables mentioned above should be considered for engineering designs. Additionally, the presented statistical data support the mechanical results, demonstrating that a variation in any of the main factors drastically impacts the response variables. Regarding the explanations for the influence of each controllable factor on the response variables, an increase in cement and lime content leads to more significant cementitious reactions/bridges in the cemented mixtures, thereby influencing the development of strength and stiffness. As for the dry unit weight and curing time, an increase in dry unit weight enhances friction mobilization between particles, while an increase in the curing period allows for the development of cementitious reactions, resulting in increased strength and stiffness of the cemented mixtures.

## 4. Conclusions

The general objective of this study is to assess the mechanical, stiffness, and microstructural characteristics of stabilized marine clay- CLW blends using the porosity-to-cement index. The study aims to fill knowledge gaps regarding the performance of these blends in geotechnical applications by examining their strength, stiffness, and microstructure. Microstructural analysis through SEM and EDX tests will provide insights into the interfacial interaction between soil, cement, and fine CLW particles and its impact on porosity reduction. Concerning the experimental and theoretical findings outlined in this research, and mindful of the study’s limitations, the following inferences can be drawn:

Through the results of mechanical strength, stiffness, and microstructure, it is possible to confirm the improvement of the fine soil through the artificially cemented soil technique as a product of the formation of a material with cementitious properties obtained from the addition of cement and CLW after 7 and 28 days of curing.

The porosity/cement index proved to be highly effective in establishing solid correlations between the mechanical strength and stiffness of the compact mixes. This compelling evidence underscores the pivotal role of this factor in the technological control of cemented soil. By employing the porosity/cement index as a reliable indicator, engineers and technicians can accurately assess and optimize the structural properties of cemented soil, thereby ensuring enhanced stability, durability, and overall performance of construction projects. This significant finding validates the importance of considering the porosity/cement index in soil cementation processes and provides valuable insights for future research and practical applications in geotechnical engineering.

The microstructure analysis of cemented soil mixtures revealed the formation of C-S-H gels and ettringite crystals when cement was added. The addition of limestone particles did not react with cement but formed a strong matrix with the foundation products. Limestone particles increased the surface area for hydration precipitation and reduced water requirements. Ettringite improved the resistance and durability of the cemented soil. The microstructure of soil−cement−CLW blends, including soil particles, cementitious matrix, and the interfacial transition zone, significantly influenced the mechanical behavior and performance of the composite. Understanding and optimizing these microstructural characteristics are crucial for achieving the desired strength, durability, and overall performance of the new geomaterial.

## Figures and Tables

**Figure 1 materials-16-04983-f001:**
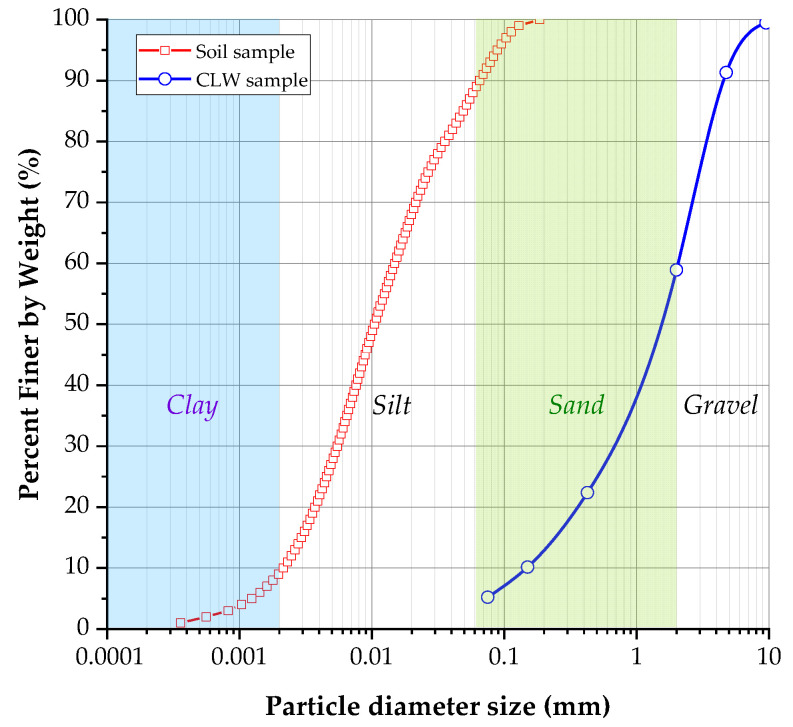
The granulometric curve of soil sample and crushed limestone waste (CLW).

**Figure 2 materials-16-04983-f002:**
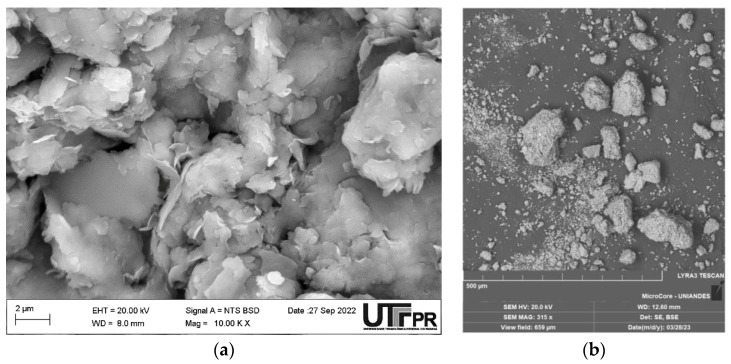
SEM images of soil sample (**a**) and CLW (**b**).

**Figure 3 materials-16-04983-f003:**
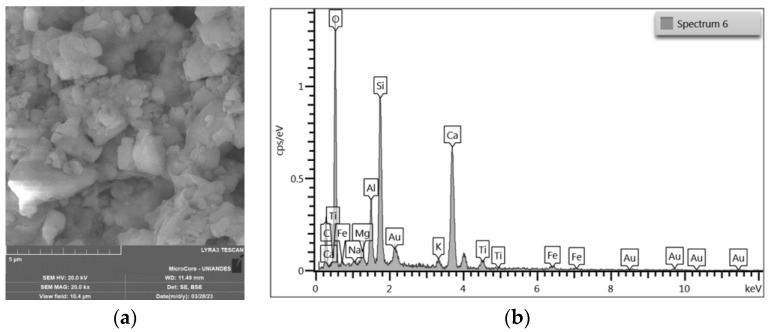
SEM-EDX of CLW. (**a**) SEM of CLW magnified at 20,000 times; (**b**) EDX chemical analysis of CLW detected region.

**Figure 4 materials-16-04983-f004:**
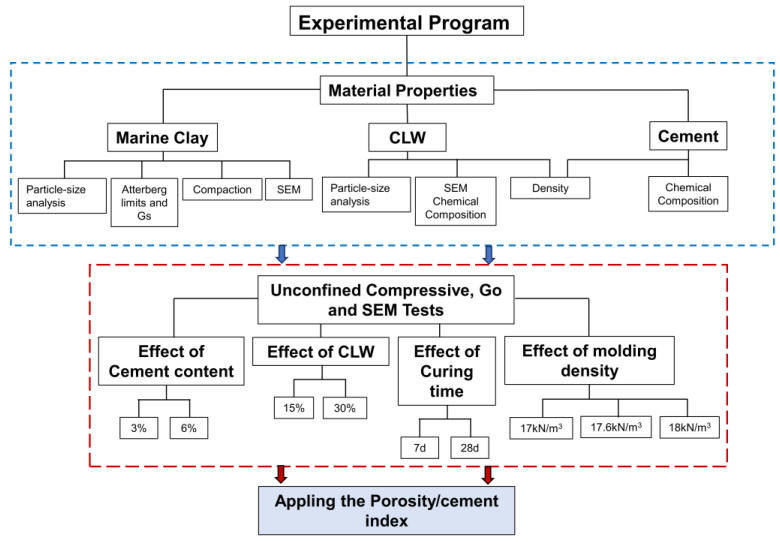
Experimental Program Flowchart.

**Figure 5 materials-16-04983-f005:**
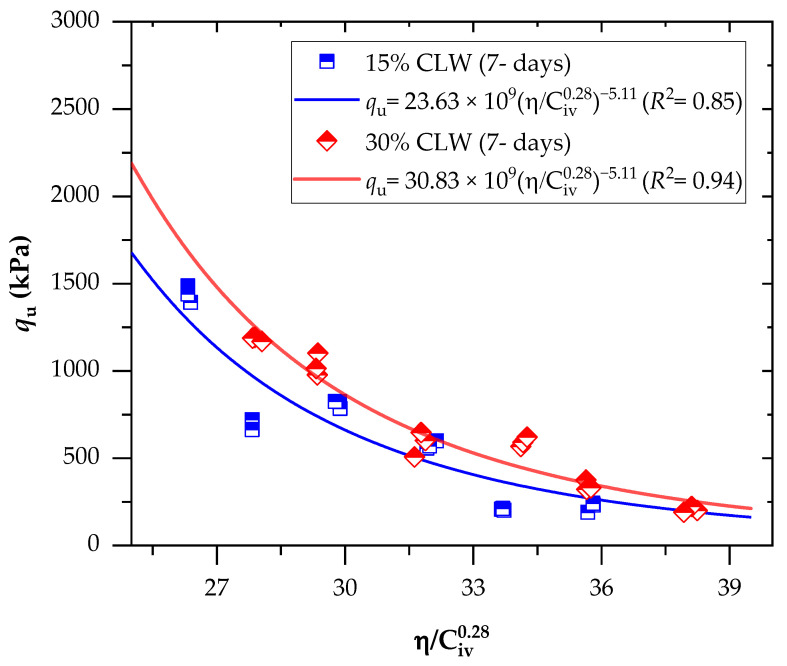
Effects of the porosity-to-cement index ($\eta /C_{iv}^{0.28}$) on the UCS of soil−CLW−cement compacted blends considering 7 days of curing.

**Figure 6 materials-16-04983-f006:**
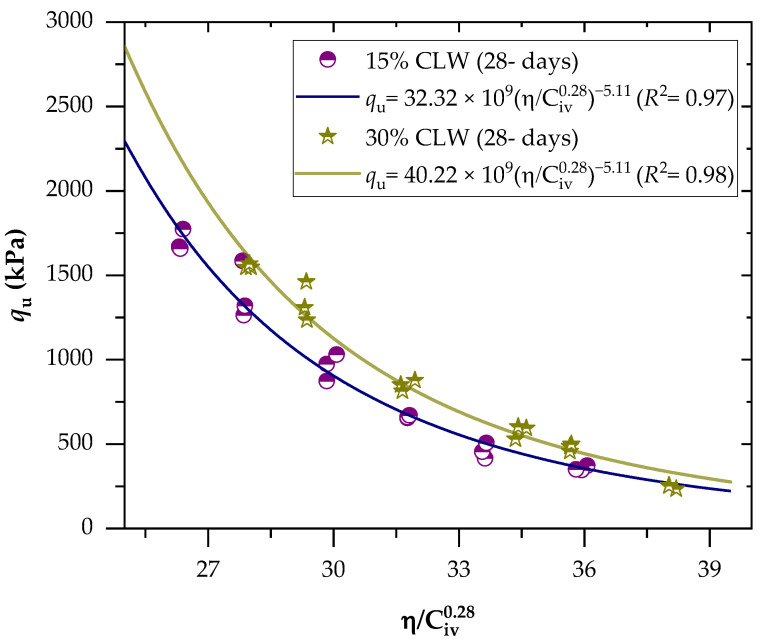
Effects of the porosity-to-cement index ($\eta /C_{iv}^{0.28}$) on the UCS of soil−CLW−cement compacted blends considering 28 days of curing.

**Figure 7 materials-16-04983-f007:**
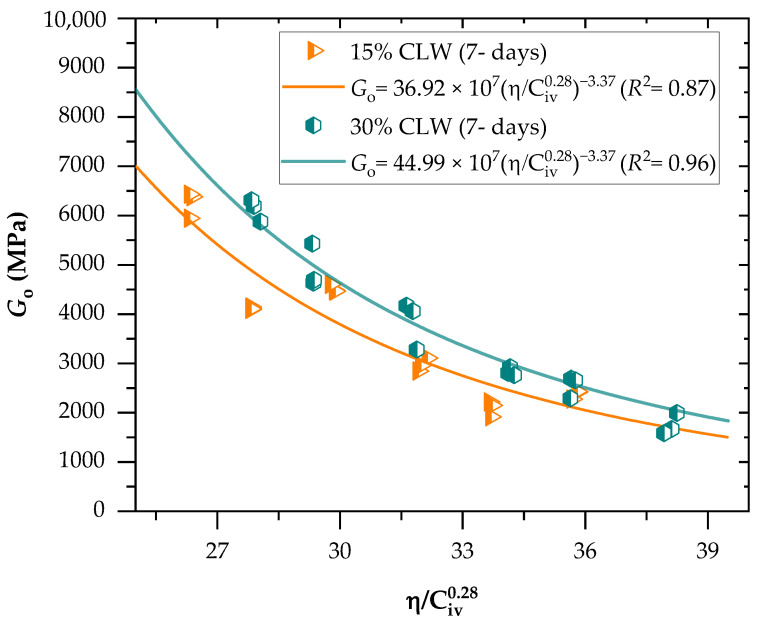
Effects of the porosity-to-cement index ($\eta /C_{iv}^{0.28}$) on the stiffness of soil−CLW−cement compacted blends considering 7 days of curing.

**Figure 8 materials-16-04983-f008:**
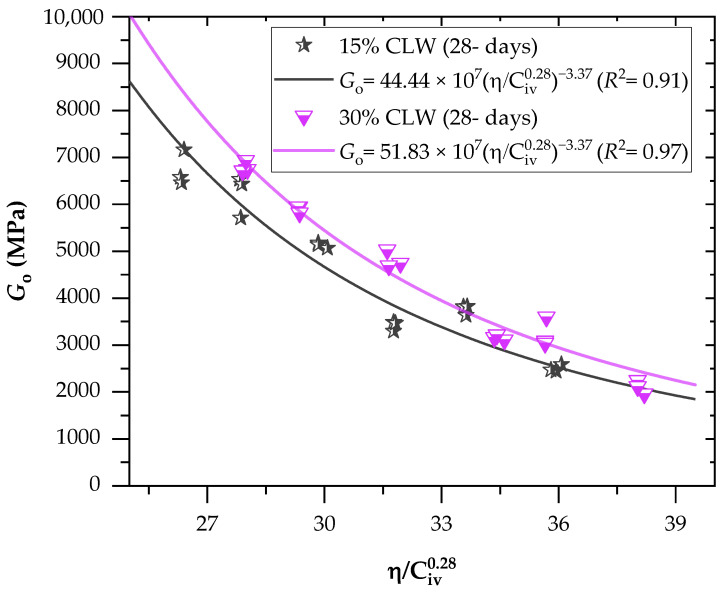
Effects of the porosity-to-cement index ($\eta /C_{iv}^{0.28}$) on the stiffness of soil−CLW−cement compacted blends considering 28 days of curing.

**Figure 9 materials-16-04983-f009:**
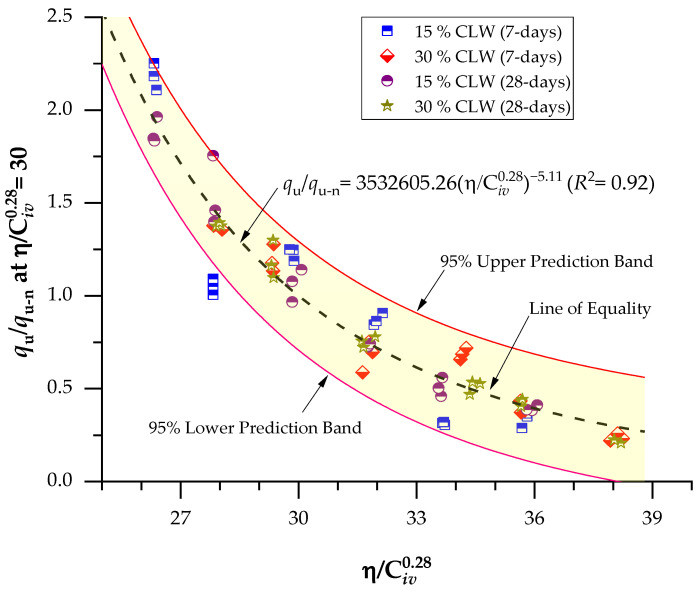
Normalization of the UCS at $\eta /C_{iv}^{0.28}=30$ considering 15–30% CLW and 7–28 days of curing.

**Figure 10 materials-16-04983-f010:**
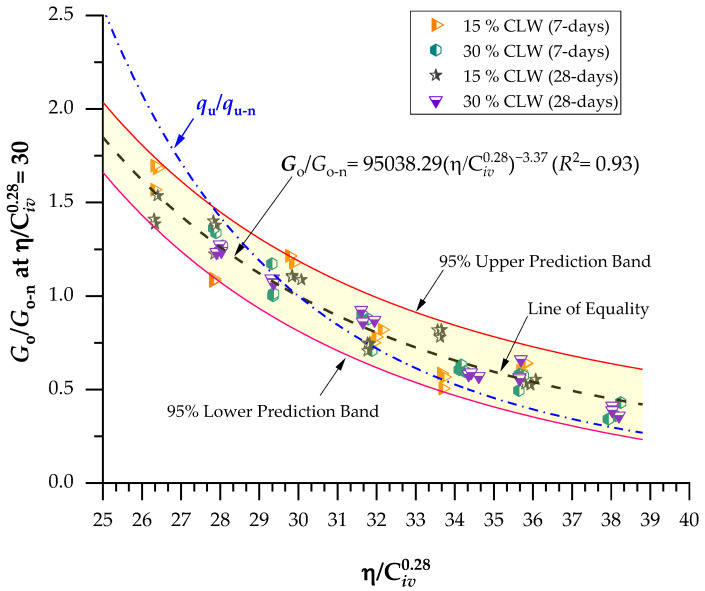
Normalization of stiffness at $\eta /C_{iv}^{0.28}=30$ considering 15–30% CLW and 7–28 days of curing.

**Figure 11 materials-16-04983-f011:**
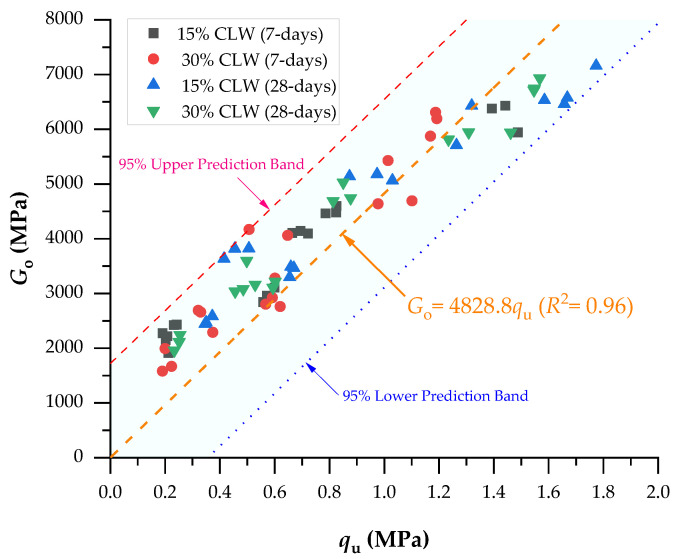
The direct relationship between the UCS and stiffness.

**Figure 12 materials-16-04983-f012:**
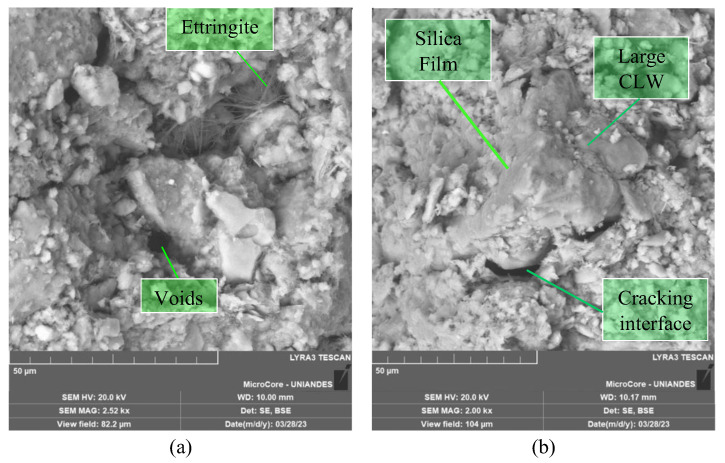
Microstructure of soil−CLW−cement compacted blend at η$/C_{iv}^{0.28}=35.6$ after 7 days of curing. (**a**) Formation of ettringite. (**b**) Unreacted CLW large particle and silicon film formation.

**Figure 13 materials-16-04983-f013:**
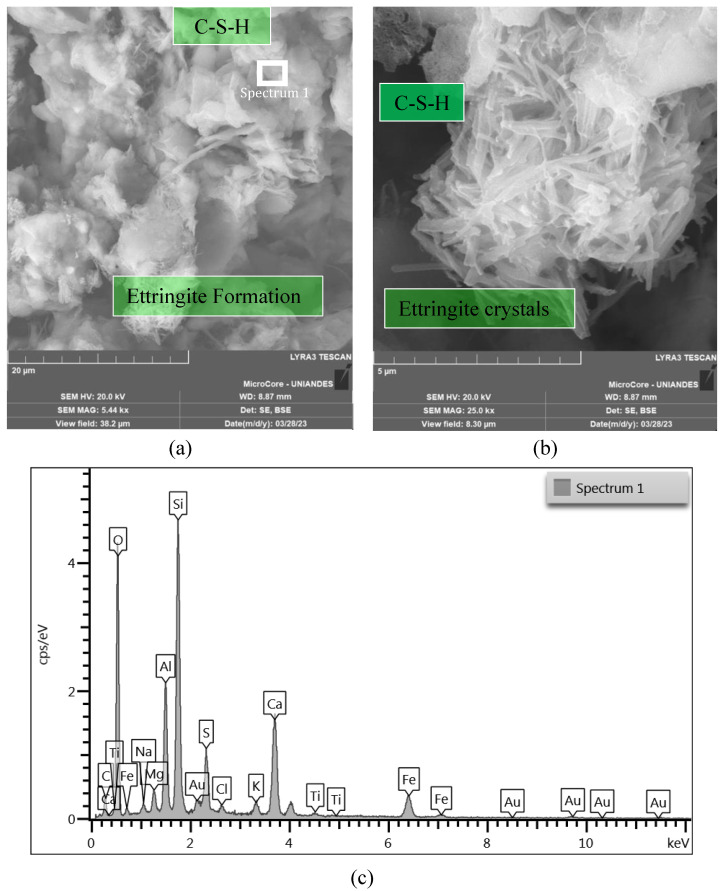
Microstructure of soil−CLW−cement compacted blend at η$/C_{iv}^{0.28}=29.5\%$ after 7 days of curing. (**a**) Loose packing of ettringite, (**b**) detail of the formed ettringite crystals. (**c**) EDX analysis on C-S-H crystal formation.

**Figure 14 materials-16-04983-f014:**
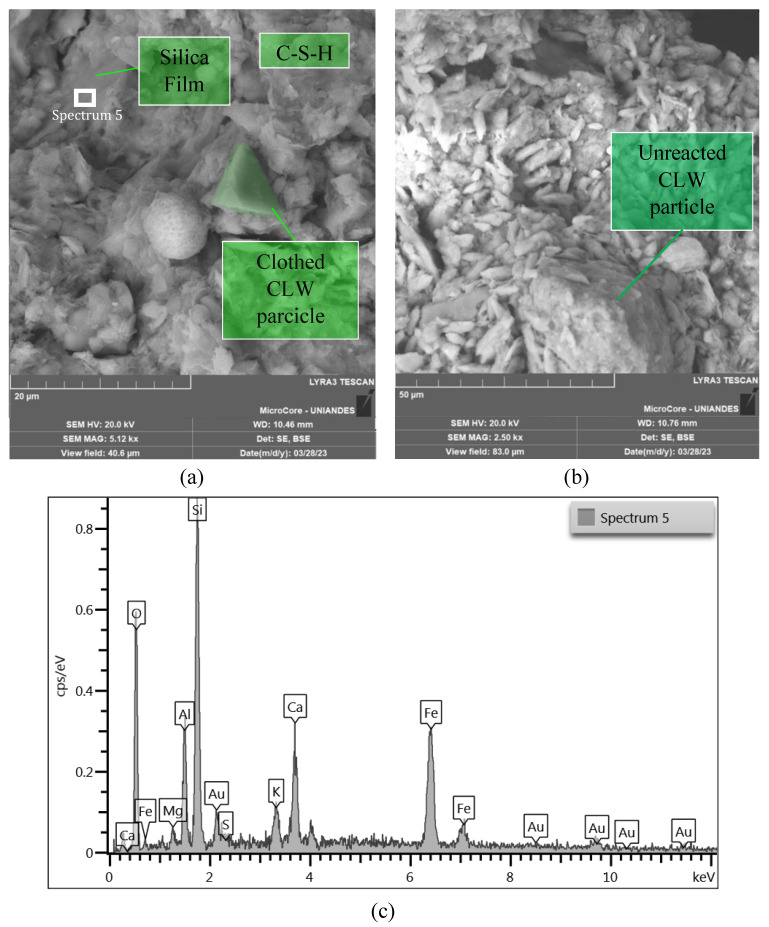
Microstructure of soil−CLW−cement compacted blend at η$/C_{iv}^{0.28}=27.8\%$ after 28 days of curing. (**a**) Silica film, unreacted CLW particle, and calcite crystal; (**b**) C-S-H formation and unreacted CLW; (**c**) EDX analysis under silica film formation.

**Table 1 materials-16-04983-t001:** Physical properties of soil sample and CLW.

Properties	Soil	CLW
LL Limit Liquid of soil, %	42.00	-
PL Plastic Limit of soil, %	26.05	-
PI Plastic Index of soil, %, (i.e., LL-PL)	15.95	-
Gravel particles (Ø-2 mm), %	0	41
Coarse sand particles size (0.6 mm-Ø-2 mm), %	0	32
Medium sand particles size (0.2 mm-Ø-0.6 mm), %	0	13
Fine sand particles size (0.06 mm-Ø-0.2 mm), %	12	14
Silt particles size (0.002 mm-Ø-0.06 mm), %	78	-
Clay particles size (Ø < 0.002 mm), %	10	-
Effective size of materials (D_10_), mm	0.0021	0.15
Mean particle diameter (D_50_), mm	0.011	1.6
Uniformity coefficient of materials (C_u_)	7.14	13.67
Coefficient of curvature of materials (C_c_)	0.96	1.59
The specific gravity of soil sample and CLW	2.80	2.52
Activity of clay, A [A = PI/(% < 0.002 mm)]	1.60	-
Color of marine of raw materials	Black	Gray
Classification of raw materials (USCS)	CL	SW

**Table 2 materials-16-04983-t002:** Chemical composition of soil sample and CLW.

Element	Soil Composition (%)	CLW Composition (%)
SiO_2_	66	9.0
Al_2_O_3_	21.7	1.3
SO_3_	5.0	-
K_2_O	3.1	-
CaO	3.0	72.4
Fe_2_O_3_	0.9	0.9
TiO_2_	0.3	-
MgO	-	2.1
Mn	-	14.3

**Table 3 materials-16-04983-t003:** ANOVA table for the UCS results.

Source	Sum of Squares	Degrees of Freedom	Mean Squares	*Z*	*p*-Value	Significance (*p*-Value < 0.05)
Cement (C)	187,370.055	1	187,370.055	96.994	<0.001	yes
CLW	200,929.215	1	200,929.215	104.013	<0.001	yes
γd	2,296,409.103	7	328,058.443	169.822	<0.001	yes
Curing time (t)	8856.464	1	8856.464	4.585	0.039	yes
C * CLW	398,023.336	1	398,023.336	206.040	<0.001	yes
C * γd	2,259,401.699	3	753,133.900	389.866	<0.001	yes
C * t	51,955.811	1	51,955.811	26.895	<0.001	yes
CLW * γd	2,446,021.981	5	489,204.396	253.241	<0.001	yes
CLW * t	161,796.778	1	161,796.778	83.756	<0.001	yes
γd * t	369,913.213	5	73,982.643	38.298	<0.001	yes
Error	75,339.212	39	1931.775			
Total	57,151,136.704	72				

* Interaction.

**Table 4 materials-16-04983-t004:** ANOVA table for the stiffness results.

Source	Sum of Squares	Degrees of Freedom	Mean Squares	*Z*	*p*-Value	Significance (*p*-Value < 0.05)
Cement (C)	52,075,561.361	1	52,075,561.361	746.458	<0.000	yes
CLW	461,084.863	1	461,084.863	6.609	<0.014	yes
γd	17,869,755.772	7	2,552,822.253	36.592	<0.000	yes
Tempo (t)	5,291,448.551	1	5,291,448.551	75.848	<0.000	yes
C * CLW	76,122.097	1	76,122.097	1.091	0.302	no
C * γd	2,278,178.994	3	759,392.998	10.885	<0.000	yes
C * t	662,158.109	1	662,158.109	9.491	<0.004	yes
CLW * γd	468,352.678	5	93,670.536	1.343	0.264	no
CLW * t	208,152.497	1	208,152.497	2.984	<0.091	no
γd * t	3,357,848.111	5	671,569.622	9.626	<0.000	yes
Error	3,139,360.982	45	69,763.577			
Total	1,384,692,258.529	72				

* Interaction.

## Data Availability

Not applicable.

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
