# Peer review of "Strength, Stiffness, and Microstructure of Stabilized Marine Clay-Crushed Limestone Waste Blends: Insight on Characterization through Porosity-to-Cement Index"

_materials, 2023, doi:10.3390/ma16144983_

Round 1
Reviewer 1 Report
This paper presents research on the characterisation of stabilised marine clay-crushed limestone waste blends using a novel approach, the porosity-to-cement index, which can contribute to the knowledge in the field and can be appreciated by peers. However, several minor issues should be addressed.
1. Research questions should be presented clearly in Section 1.
2. A workflow should be made to explain the experiment step in Section 2.
3. The author mentioned a novel approach. However, the author should compare the approach with the existing approaches in Section 2 to emphasise the novelty.
4. Necessary references should be fully cited in Section 2 to support the specimen preparation and experiment conditions.
5. 5. Conclusion should be changed to 4. Conclusion. The number is wrong.
6. Line 423, Table 4 and Table 5 should be changed to Table 3 and Table 4, respectively. The numbers are wrong.
7. More detailed explanation of Table 3 and Table 4 should be supplemented in Section 3.3.
Author Response
Please, see the Attached.

Reviewer 2 Report
This paper presents a study of the mechanical behaviour of stabilised soil-cement mixtures with different porosity/cement indexes and curing times. The authors aimed to develop a single mathematical correlation capable of predicting the mechanical behaviour of the stabilised material. The study found that the porosity/cement index is an important factor to consider in soil cementation processes and that the microstructure of soil-cement-CLW blends significantly influences the mechanical behaviour and performance of the composite. The addition of limestone particles did not react with the cement, but formed a strong matrix with the foundation products, and ettringite improved the strength and durability of the cemented soil. The authors suggest that the mechanical response of soil-cement mixtures using the same cement can be standardised by taking into account various predetermined factors such as curing time and temperature. The study was funded by the Fundación Universitaria Los Libertadores-Colombia (FULL) (Project N° ING-08-23) and the authors declare no conflict of interest.
The paper is well written but some important aspects such as the hypothesis (and in its absence a general aim) and sub-aims to prove your hypothesis are missing.
In addition, section 3.2. should be restructured by moving the general equation to the experimental part as it is presented in the results and discussion section. The authors should indicate how they have obtained this equation or explain it as they do in the section I have indicated.
Although the corrections may seem major, they are not, and I believe that their work will be improved.
Once these changes have been made, the paper could be a candidate for publication in the journal Materials.
Other remarks to be considered by the authors are the following:
Line 67: Put a space between the link and the text. Same comment for line 72.
Line 81: Why are these marine sediments classified as waste? What is their origin?
Last paragraph of the introduction: the authors should state a hypothesis, not an intention of what they want to do. The hypothesis would answer their research question, which would be the overall aim, and the authors should also reflect on the steps or variables they will use to test their hypothesis.
Units throughout the text should be formatted as g cm^-3, not g/cm^3. Apply the correct formatting throughout the document.
Line 130: Decimals are excessive given the precision of the technique, authors should use only one decimal place.
Try not to cut Table 1 between the two pages (as the formatting of the article is done by the authors). If necessary, include the caption and add Table 1 (continued). Same comment for Table 2.
In Table 2 I suggest to use only one decimal place, the uncertainty of these measurements is very high and therefore 2 significant figures are too many.
Line 162 has a space between 50 and mm. Also for 0.01g.
Line 178: write 2 mm.
Line 204: indicate which software the authors are referring to.
Line 224: If expression 1 has not been determined by the authors, it should be included in the experimental part.
Section 3.1. The authors refer to the term correlation, when in fact they check the goodness of fit with the coefficient of determination, which measures the degree of dispersion explained by the selected model. Obviously, the selected models explain well the dispersion of their experimental points, with percentages higher than 80% in all cases. If you want to talk about correlations, you should determine them with any statistical program such as StatGraphics...
Line 371: try to keep the figure captions on the same page.
Figures 11, 12 and 13, unify the font in the figures and make it larger so that it is easy to read.
Tables 3 and 4 are unnecessary or not easy to understand, it is not clear to me how the ANOVA of the factors was done. Obviously all the p-values below the 0.05 significance level allow the null hypothesis to be accepted, but it is not really clear to me what they are comparing.
Change the format of the conclusions. It is better to put the overall aim (which was not stated in the introduction) in the first paragraph and then the sub aims if they have been achieved and have helped to test your hypothesis. This would be fine, but without enumeration.
The authors do not refer to any work published in the material, they could try to look for some works related to the topic.
Conclusions should be numbered 4.
Author Response
Please, see the Attached.
